# Parallel Processing of Sensor Data in a Distributed Rules Engine Environment through Clustering and Data Flow Reconfiguration

**DOI:** 10.3390/s23031543

**Published:** 2023-01-31

**Authors:** Adrian Alexandrescu

**Affiliations:** Department of Computer Science and Engineering, Faculty of Automatic Control and Computer Engineering, Gheorghe Asachi Technical University of Iaşi, Str. Prof. dr. doc. Dimitrie Mangeron, nr. 27, 700050 Iași, Romania; adrian.alexandrescu@academic.tuiasi.ro or aalexandrescu@tuiasi.ro

**Keywords:** parallel processing, smart city, sensor, rules engine, k-means clustering, genetic algorithm, sensor network, clustering, cloud computing

## Abstract

An emerging reality is the development of smart buildings and cities, which improve residents’ comfort. These environments employ multiple sensor networks, whose data must be acquired and processed in real time by multiple rule engines, which trigger events that enable specific actuators. The problem is how to handle those data in a scalable manner by using multiple processing instances to maximize the system throughput. This paper considers the types of sensors that are used in these scenarios and proposes a model for abstracting the information flow as a weighted dependency graph. Two parallel computing methods are then proposed for obtaining an efficient data flow: a variation of the parallel k-means clustering algorithm and a custom genetic algorithm. Simulation results show that the two proposed flow reconfiguration algorithms reduce the rule processing times and provide an efficient solution for increasing the scalability of the considered environment. Another aspect being discussed is using an open-source cloud solution to manage the system and how to use the two algorithms to increase efficiency. These methods allow for a seamless increase in the number of sensors in the environment by making smart use of the available resources.

## 1. Introduction

As technological development spreads across the world, there is a tendency to automatize and remove the human element from many aspects of everyday life, from automated production lines to solutions based on artificial intelligence such as self-driving cars or generating human language.

When it comes to improving the day-to-day lives of people and their respective communities, the concept of smart-entities emerges, which make use of the Internet of Things (IoT). Examples of entities are homes, buildings or cities. The term IoT started in 1999 and involved interconnected entities by means of radio frequency identification technology. Nowadays, the IoT is characterized by concepts such as wireless sensor networks (WSNs), identifiable devices and actuators, cloud computing or low energy [1]. A wireless sensor network (WSN) represents a network of spatially dispersed sensors that monitor various aspects of the environment in which it is deployed.

The Internet of Things concept is the cornerstone of smart entities. A simple form of smart entity is a smart home, in which basic heating, ventilation and air conditioning (HVAC) elements are automatically managed. HVAC implies the use of different technologies to control heating, ventilation and air conditioning in the home or other buildings. The next step is the smart building [2,3], in which the building’s functions are automatically controlled to improve the lives of the residents while being cost and energy efficient. Another step forward is smart cities [4,5], which take the concepts of comfort and efficiency even further by focusing on sustainability, connecting citizens to various public services, managing traffic and providing utilities. There are also spin-offs to these concepts such as smart villages for developing and introducing smart technologies to rural areas [6] and smart regions [7], which focus on connecting smart cities while having in mind a regional strategy for “promoting smartness”. Another aspect of smart entities is the use of artificial intelligence to identify patterns and make decisions.

Regardless of the goal of smart entities, a very high amount of sensors is required. Those sensors produce large volumes of information that need to be handled, in most cases, nearly in real time.

There are several surveys regarding the different types of sensors employed in an IoT environment [8,9,10,11]. The main sensor category types that can be deployed in an IoT environment and smart cities include ambient, motion, vital, identification, positioning, entity detection or presence, interaction, acoustic, hydraulic, force or load, vibration or chemical sensors.

A key element of processing sensor data is the use of a rule engine. A rule engine allows the definition of multiple rules that activate when a condition is met. The condition is usually a comparison between values obtained from sensors and other fixed data. Thus, the activation can be triggered depending on if a sensor reading changes. If there are multiple sensors in the system, then there are also multiple rules that handle the data. All this information must be processed as fast as possible.

The current paper focuses on handling large volumes of data from sensors by efficiently using a limited amount of computing resources. The considered environment comprises multiple sensors, sensor networks, gateways, data storage solutions and computing instances. The main goal is to determine a strategy for distributing the sensor data processing among the available computing resources.

The approach proposed in the current paper focuses on processing raw sensor data in a central point with distributed computing capabilities. Multiple gateways can continue to exist in the system, but their role is just to filter noise and forward the data. Two methods are proposed for determining the data flow strategy: one based on the k-means clustering algorithm and a variation of a genetic algorithm.

The main contributions and novelty that the current paper proposes are summarized as follows:A sensor data processing architecture in which raw sensor data from a large number of different sensors is efficiently processed in a distributed computing environment;An abstraction of sensor data processing by using rule engines, which allows data flow configuration by means of algorithms that analyze communication patterns;Two methods for streamlining data flow and improving data processing by creating clusters. where these algorithms process the information by taking into account the number of sensors for each rule and the data volume from each sensor. The two methods are the following:
-An adaptation of the k-means clustering algorithm;-A genetic algorithm with a complex fitness function based on two desired criteria;A solution for cloud deployment while ensuring scalability and adaptation to the sensor rule particularities of the system.

The remainder of this paper is organized as follows. Section 2 discusses how the proposed architecture and the two proposed methods compare to existing solutions for the considered topic. Section 3 presents the considered context as well as the proposed goal, the rules engine, the sensor rule abstraction, the performance metrics that were used, the two proposed clustering methods and the solution to deploying the system for a cloud solution. Section 4 describes the simulation set-up and the experimental results. Finally, Section 5 showcases the importance to the proposed methods and how they perform when considering various other aspects of the environment. Another discussion in this section is related to scalability and managing a dynamic environment.

## 2. Related Work

The related work presented in this section is concentrated on the main aspects of the proposed research: rule engines, sensor networks, data flow reconfiguration and clustering.

Existing rule engines focus on the representation of the facts that are used in handling sensor information. This is performed as a way to increase the efficiency of processing a large amount of sensor data. In [12], the authors extracted atomic events from sensor data and proposed a scheme that minimized the rule-matching overhead. Another related piece of research is [13], where the authors proposed a rule engine which allowed flexible rule strategies. In terms of smart rules engines, in [14,15], the authors presented a solution based on fuzzy logic for processing information. The research presented in the current paper uses the rule engine from [16], mainly due to its simplicity. The way the proposed system is designed, it allows any of the aforementioned rule engines to be used as long as the time it takes to process the sensor data is directly dependent on the volume of received sensor data.

There is existing research regarding various aspects of the considered environment, but none of it focuses on optimizing the data flow in the system in both a centralized manner (by pooling data from all of the sensors) and a decentralized manner (by distributing the processing to multiple instances). The advantage is that multiple gateways can be used to communicate with specific sensor networks, and they each can provide local processing while providing the results up the hierarchical chain, perhaps to a central cloud. This method has the disadvantage of requiring computational capabilities for each gateway, and some relevant data may also be overlooked if they are processed locally. Those data may be relevant in the global scope.

In terms of environments with multiple connecting gateway systems and wireless sensor networks, the current research topics cover a wide area. An example of handling communication in hierarchical wireless sensor networks with multiple gateways is presented in [17]. However, in that research, there is a method proposed for solving the authentication problem that is not focused on improving information flow but rather on securing access to resources.

Another aspect which is discussed in the current paper is related to data flow reconfiguration. Although the main focus is obtaining a very good configuration from the start, Section 5 of this paper tackles the dynamic reconfiguration scenario. In terms of network configuration, the authors of [18] proposed a platform for fast sensor network prototyping. Regarding dynamic network reconfiguration, there is existing research in this field [19] and also comparisons between existing methods [20,21]. None of that research can be directly applied to the considered environment of multiple rule engines that have to process the sensor data efficiently.

Much research regarding the considered environment focuses on improving energy efficiency. In [22], two methods were proposed for optimizing energy consumption in a WSN for monitoring a smart city. Even though the k-means clustering algorithm was employed, the features used for the algorithm and the overall scope differ from the research proposed in the current paper. Another paper involving clustering, gateways and sensors used in a different scope is [23]. The authors proposed that there is a solution for improving energy consumption in a wireless body area network used for patient monitoring. The used clustering method selects cluster heads based on the residual energy of the cluster nodes. Other papers used hierarchical clustering [24,25] and other techniques [26] to determine the optimal number of clusters for minimizing energy consumption. While those papers used clustering to achieve their proposed goals, none of them focused on the particular problem considered in the current paper.

The research in [27] used k-means clustering in order to enhance privacy for data clustering of information from an IoT environment. The idea was centered on improving privacy and not on increasing the efficiency, as is the case in this proposed research. Another paper involving clustering in WSNs is [28], but there the authors considered multi-hop routing trees and reduced the total energy consumption. In [29], the authors focused again on energy consumption and the distribution of cluster heads. These papers had more features available for consideration through the k-means algorithm, which led to an increase in the efficiency of using that algorithm in solving a specific problem. Other related research includes parallel clustering on low-power devices in edge computing environments [30] or methods for delivering sensor data and storing them in an IoT environment [31,32].

## 3. Proposed Architecture and Methods

### 3.1. Considered Context and Proposed Goal

Smart buildings and smart cities employ thousands and even tens of thousands of sensors to monitor various parameters and to interact with the environment based on specific rules. The data from those sensors need to be processed nearly in real time.

Three issues arise: how to obtain the data from the sensors, where to store them and how to properly process said data. In terms of obtaining data from sensors, a good approach is the one from [33], in which there are bridge and driver processes that gather data from one or more sensors belonging to a specific communication protocol (e.g., Z-Wave, ZigBee or other standardized APIs) and filter data noise. Storing the data can be accomplished in two main ways: message-queuing services (e.g., MQTT broker, RabbitMQ or a cloud queue solution) or databases (e.g., a data store or data lake, usually stored in a public or private cloud). Regardless of the storage solution, the sensor information must be efficiently and swiftly processed in order to take specific actions, or large amounts of data can be processed to find certain patterns. Either way, there is the need for a gateway component to use the acquired data and perform some computations and processing.

The aforementioned approaches work well for smart home and smart building systems, but for smart cities and smart regions, the incoming data need some sort of distributed computing in order to be properly processed. Figure 1 shows two approaches for handling the data flow and processing.

The first architecture (Figure 1a) is the traditional approach, in which local gateways process the information from sensors. That information is used to make decisions and, from time to time, to send relevant data to a central node or gateway for storage and further processing. This a good approach, but it has two disadvantages. First, the central node does not have access to all the local data, and it can be less efficient at finding patterns. Secondly, each gateway is an extra computing component and overhead to the entire system. On the other hand, it makes quick local decisions, but without the overall picture, those decisions might not be the most efficient ones.

The second architecture (Figure 1b) starts from the premise that all the sensor data are directly sent to the central distributed storage (e.g., a distributed message queue service or massive distributed data store). Having a central distributed data-processing component allows having an overview of the entire environment and triggering local decisions based on the general context. Another advantage is having fewer overall data processing instances, because one instance can process data from multiple and various sources. The research presented in this paper focuses on the second architecture model, and the two proposed solutions aim to improve the data processing performed by each instance.

### 3.2. Data Processing and Rule Engines

Processing sensor data consists of having data from specific different sensors as input, computing those data by some means and triggering one or more actions if needed.

Basic processing can be achieved by employing rule engines [16]. Each piece of data passes though the rule engine, which looks at all the encompassing rules, determines which ones trigger (if any) and takes the associated action(s). The rules can be described in JavaScript Object Notation (JSON) format as shown in Listing 1. JSON is a heavily used text-based method of representing structured data in a both human- and machine-readable format. In this example for a home room, a rule is set to trigger turning on the air conditioning unit in dehumidify mode if the humidity in the room is over 55% and the temperature is over 26 degrees Celsius. The condition for this rule is checked each time the humidity or temperature in the room changes. Slight variations will not trigger the rule because one of the roles of the sensor data acquisition component is to send the new data only if the change in value is above a certain increment.

**Listing 1.** Example of a rule for HVAC in JSON string format based on the rule engine
described in [16].{  "name": "humidity-control-on",  "description": "If the humidity in the room is over 55% and the temperature is over 26 degrees Celsius,          ↪ then start the AC and set it to dehumidify",  "priority": 3,  "condition": "${home-room-1/hvac/humidity}{value} > 55 and ${1/hvac/temperature}{value} > 26",  "actions": [      "${home-room-1/hvac/ac}{value}{on}",      "${home-room-1/hvac/ac}{mode}{dehumidify}"  ]}

The above example is a trivial one, but the rule engine can be used for complex rules that depend on different types of sensors and even on other rules. Anything that has a unique identifier in the system (e.g., home-room-1/hvac/humidity) can be referenced in both the condition and the action properties of the rule. The rules are not exclusive to HVAC or home monitoring and can be employed for monitoring traffic, for automatic street lightning or even for triggering facial recognition algorithms when using CCTV. The latter requires a component that processes the images, filters them and, from the rule engine point of view, acts as a data-producing sensor. Nonetheless, this is beyond the scope of the current paper.

To summarize, the considered architecture employs a rule engine for each computing instance, and each rule engine manages a set of rules whose conditions need to be checked, depending on the incoming sensor data.

### 3.3. Sensor: Rule Abstraction

A high amount of computational power is required when having a large amount of sensors which produce data and many rules that need to be checked, depending on the received data. As previously established, the proposed solution is to distribute the load among computing instances. Therefore, each instance will handle a distinct set of rules, and each rule needs the data from specific sensors. It does not make sense to check the rules from all the instances once a new piece of sensor data enters the system; rather, only the rules that involve that sensor should be checked. Only the instances with the involved rules will ever receive data from that sensor.

A rule can be seen as a generic check if certain conditions are met, and if that conditional expression is true, then one or more actions are triggered. This concept can be extended to include the possibility of triggering more complex processing if specific sensor values satisfy certain conditions. Each sensor produces, by means of the data acquisition component, a variable amount of data depending on the sensor type and the environment conditions. The system can be monitored in order to determine the approximate amount of produced data for each sensor in a time period (e.g., hour, day, month and year).

Regardless of the sensor type and the acquired data format, the information that gets transmitted by means of WiFi, Bluetooth or other methods is basically an array of bytes. In order to abstract the volume of the acquired and transmitted sensor data, the data quantities are in generic data units (d.u.). These units represent the amount of information from one sensor during a specific period of time.

An example of connections between sensors and rules is presented in Figure 2. There are six sensors ($S_{1}$–$S_{6}$) and four rules ($R_{1}$–$R_{4}$), and each edge represents the volume of information that gets sent from a sensor to a rule. The volume of information is a numerical value expressed in generic data units (d.u.). In a real scenario, this value represents the number of bytes sent from a sensor to a rule. Each rule needs data from specific sensors (e.g., rule $R_{1}$ requires 12 d.u. from $S_{1}$ and 10 d.u. from $S_{2}$). Therefore, rules need data from multiple sensors, and sensors send data to multiple rules.

For simplicity’s sake, the scenario considered in the proposed paper implies that rules cannot send data to other rules, but this can easily be made possible if certain rules are set to also act as sensors (i.e., produce data). For each rule that produces data, a virtual sensor would be created in the system to simulate this. If a cycle is created in the resulting graph by considering this scenario, then the clusterization approaches proposed in this paper are still valid as long as the volume of information being sent from one rule to another is quantifiable. By knowing how many data are being sent in this case prior to running the algorithms, the proposed solutions work perfectly.

As mentioned earlier, the problem statement is that there is a limited amount of computing instances available. Each computing instance must process a group of rules so that the information that is being sent to that instance is minimal. Therefore, the goal is to somehow group the rules into clusters so that the information is being sent efficiently to the clusters. Each cluster of rules is then the responsibility of a single computing instance.

Figure 3 and Figure 4 show two of the possible different clusters that can be formed. Both figures show two clusters, but each cluster has a different composition and, consequently, will have a different amount of data volume that needs to be processed. In the first case (Figure 3), cluster $C_{1}$ encompasses rules $R_{1}$ and $R_{3}$ and receives a total of 31 d.u. from sensors $S_{1}$, $S_{2}$, $S_{3}$ and $S_{4}$, while cluster $C_{2}$ encompasses rules $R_{2}$ and $R_{4}$ and receives a total of 27 d.u. from sensors $S_{1}$, $S_{4}$, $S_{5}$ and $S_{6}$. In the second case (Figure 4), cluster $C_{1}$ encompasses rules $R_{1}$, $R_{2}$ and $R_{3}$ and receives a total of 38 d.u. from sensors $S_{1}$, $S_{2}$, $S_{3}$, $S_{4}$ and $S_{5}$, while cluster $C_{2}$ encompasses rules $R_{4}$, $S_{5}$ and $R_{6}$ and receives a total of 22 d.u. from sensors $S_{4}$, $S_{5}$ and $S_{6}$.

When a sensor is removed from the system, the corresponding rules need to be updated to account for this. Removing just a single sensor in the context of tens of thousands of sensors will have an insignificant impact on the system’s performance. However, if multiple sensor networks are removed, then one of the two proposed methods has to run to reconfigure the data flow. When a single sensor and the associated rules are added to the system, those rules can be assigned to the cluster with the smallest load. When adding multiple sensor networks to the system, the clustering process needs to trigger again. As it stands, the clustering method might perform a complete reconfiguration, which might trigger multiple rule migrations from one instance to another. Depending on the complexity of the migration, this might not be desired. On the other hand, this massive migration might provide a much better processing time in the long run. Therefore, when considering the dynamism of the proposed system, the scalability is highly dependent on the moments when the reconfiguration is triggered and on the impact of the rule migrations. The best way to achieve high scalability is achieved in two steps. First, when sensors are added to the system, they are assigned to the least loaded instance. Secondly, a job scheduler needs to be set up so that the clustering algorithms periodically reconfigure the data flow. The reconfiguration implies transferring rules in JSON format from one instance to another, which can be performed in a timely manner. The sensors need to know to which instance they should send the data. This problem can be easily solved by using a gateway or load balancer. The sensors send the data to the gateway instance, which then redirects it to the corresponding instance.

The way that the clusters are formed can have a significant impact on the performance of the system. Therefore, performance metrics have to be set in order to quantify the quality of the clustering.

### 3.4. Performance Metrics

The main performance indicator is the speed with which the computing instances process the data. To keep the issue as simple and clear as possible, this research assumes that the time it takes to process data units from each sensor is directly proportional to the amount of data units that the instance receives. If, in reality, this is not the case, then the data unit value from each sensor can be multiplied by a factor depending on each specific sensor. Therefore, the problem becomes reduced to the equivalent of having normalized the volume of information and the time it takes to process it to generic units (i.e., data units), as is discussed further in this paper. The problem could be further reduced to a task mapping problem, but the situation is different from the standard approach because of the extra layer introduced by having rules. Another reason for this is because sensors should not send data to instances regardless of what rules are processing the information.

In terms of performance metrics, the goal is to process information as fast as possible. More data to be processed equals more time required to handle those data. When looking at the considered problem as a static task scheduling issue, where a task is considered to be the processing of a piece of data, the goal is to minimize the makespan (i.e., the time it takes for all instances to finish processing the assigned tasks) [34]. Because the environment is a dynamic one, the quality of the solution to the considered problem cannot be accurately evaluated. Using the makespan is a good performance metric in a static task allocation that uses no heuristics, but it has the disadvantage that certain instances are not used to their full potential. The makespan value is given by the instance that finishes the allocated tasks last, and there can be instances that finish much earlier.

Given the nature of the dynamic environment, static task allocation methods cannot be used, and other heuristics must be employed. The makespan is still useful in this case, but a better approach may be to consider balancing the load so both metrics must be taken into account. On the one hand, the goal is to finish processing as fast as possible, and on the other hand, the goal is to keep the load balanced among the instances.

Considering the example from Figure 3, the makespan was 31 d.u., and the load imbalance (i.e., the difference between the instance that finished latest and the instance that finished earliest) was 4 d.u. For the example in Figure 4, the makespan was 38 d.u., and the load imbalance was 16 d.u. Low values of both the makespan and the load imbalance are desired so the first clustering example is better from both points of view. Intuitively, the makespan is a good enough metric, but tests have shown that using the load balance as a criterion for certain algorithms can provide good results as well.

### 3.5. Problem and Solution Representation

The problem that needs to be solved is how to cluster the rules together so that each cluster requires the least amount of data to pass through the associated rules. The two proposed algorithms that solve this problem receive as input a graph similar to the one from Figure 2, represented by a two-dimensional array (*A*). The rows represent *n* sensors (*S*), the columns represent *m* rules *R*, and $A_{ij}$ represents the volume of data in data units (d.u.) which the sensor $S_{i}$ sends to rule $R_{j}$ for processing. Therefore, the goal is to find *p* clusters *C*, where $C_{k}\subset R_{j},j\in \left\{1..m\right\}$ and ∀k∈{1..p},∀l∈{1..p},k≠l, $∄\;C_{k}\subset R_{j}\;\mathrm{and}\;C_{l}\subset R_{j},\forall j\in \left\{1..m\right\}$.

The solution consists of a list of rules that are associated to each cluster. A rule can be assigned to only one cluster. A simple representation is using an array in which each position is associated with a rule. The value at that position represents the cluster number to which that rule belongs.

The available solution space for the considered problem is very large. The number of possible solutions is $p^{m}$, because each of the *m* rules can be associated to one of the *p* clusters. For example, if there are 10,000 rules and 36 clusters, then it would take more than a lifetime to generate all the solutions. Each time a solution is generated, it has to be evaluated, and this implies many computations. All those computations also depend on the number of sensors in the system. It is impossible to develop an algorithm that offers the best solution in a timely manner. Genetic algorithms are a good method for solving problems in which good enough solutions need to be obtained in a decent amount of time while looking at a large solution space. A drawback of genetic algorithms is the execution time. Methods that obtain faster results are traditional clustering algorithms. Clustering algorithms usually require that the metric is some sort of distance that has to be computed. The characteristics of the considered environment do not make that distance straightforward to compute. Therefore, the method that was chosen in this paper was k-means clustering because it is widely used in existing research, and its simpler form allows an easier adaptation for solving the problem.

The criteria used for grouping the rules into clusters depend on the approach. For the k-means algorithm, the parameters that can be considered are limited due to the nature of the problem. Because the way that the rule engine processes the data is considered a black box, all that remains is focusing on the data that are being transmitted. The two parameters that are taken into consideration are the number of pieces of data that are being sent by each sensor and the total volume of information that is being transmitted by a sensor to each rule. When using the genetic algorithm approach, the focus is on the end goals of minimizing the makespan and the load imbalance. This method generates multiple solutions and tweaks the resulting chromosome. Therefore, the fitness function considers the data volumes and determines the quality of a solution.

### 3.6. K-Means Clustering Approach

The k-means clustering algorithm is a good approach for grouping together items that have similar features. The basic steps of the algorithm adapted to the considered problem are as follows:Choose the number of desired clusters (*p*).Randomly choose *p* centroids (each one represents the center of the cluster).For each rule, perform the following steps:
Compute the distance to each centroid.Find the closest centroid.Assign the rule to the cluster corresponding to that centroid.For each cluster, perform the following step:
Compute the new centroids as the mean of all the rule features assigned to that cluster.Go to step 3 until there are no changes compared to the previous iteration or a specific number of iterations has passed.

Usually, this method uses the Euclidean distance for determining the closest centroid, but there are similar approaches available [35]. Regardless of the method for computing the distance, certain numerically quantifiable features need to be considered. Due to the complexity of the considered problem, modeling the input data to be suitable for the k-means method proves quite difficult. After considering a few variations, the best results were obtained by considering for each rule two features: the number of sensors that send data to that rule and the total volume of data that is being sent to that rule by all the sensors. These two features are used as metrics when computing the distance between each rule and each centroid.

This method of clustering does not help in providing a low makespan or even a balanced load; instead, it helps group together rules with similar features. To obtain a fairly well-balanced load, a second part was added to the proposed solution which split the clusters based on their size. The cluster size is represented by the total data units that are required by the encompassing rules. The novel proposed k-means-based algorithm is as follows:Choose a much lower number (e.g., q=4) than the number of desired clusters (*p*).Create *q* clusters with the previously described k-means algorithm.Compute each cluster’s size.Split evenly the larger clusters so that each resulting cluster is around the same size.Recompute each cluster’s size.Split the clusters again or combine clusters depending on the desired number of clusters while recomputing each cluster’s size if the cluster composition changed.

The advantages of this approach are that it is fast, it can be easily parallelized [36] and it provides a better alternative to assigning rules to clusters randomly. The downside of this method is that it may prove to be difficult to obtain a specific number of clusters. This approach is better suited if there is not a fixed number of clusters to be obtained.

### 3.7. Genetic Algorithm Approach

Genetic algorithms provide good solutions in a very large solution search space at the expense of possibly not finding the best solution but rather a good enough one. There are two key elements when it comes to the genetic algorithm (GA): the candidate solution representation and the fitness evaluation function. The GA starts with a population made out of individuals (chromosomes), and each chromosome represents a candidate solution. The proposed solution representation is straightforward. Each gene represents a specific cluster while its locus represents the rule, and the alleles are the possible clusters that can encompass the rules. There are two fitness functions being considered: the makespan and the load balance, as discussed in Section 3.4. The proposed genetic algorithm solution with the chosen parameter values and functions is as follows:Use the input data and initialize the GA parameters, such as in the following example:
Population size: 200.Crossover probability: 0.7.Mutation probability: 0.5.Elitism percent: 0.05.Maximum iteration count: 1000.Maximum “no change” iteration count: 300.Randomly generate the initial population.Evaluate the population using the custom fitness evaluator.While the stop condition is not met, perform the following steps:
(a)Generate a new population of the same size by going through the following steps multiple times:
Select two chromosomes using roulette wheel selection.Possibly a apply one-point crossover to each selected chromosome.Possibly apply a random allele switch mutation to each selected chromosome.Add the two resulting chromosomes to the new population.(b)Apply elitism, under which the best chromosomes from the previous population are kept in the new population.(c)Compute the fitness of each chromosome from the new population.Output the solution represented by the chromosome with the best fitness.

All the chosen parameter values and all the chosen functions (i.e., selection, crossover and mutation) are based on the existing literature regarding this topic [37,38] and on previous experiments with genetic algorithms performed by the author [39,40,41]. There is one exception regarding the mutation probability parameter, which is usually lower. Tests have shown that for the considered problem, a higher mutation probability yields better results.

The complexity of the proposed genetic algorithm is given by the fitness evaluation function, which has to take into account the sensor data needed for each created cluster. This is somewhat computationally intensive, and a caching mechanism is used so as to not recompute the fitness of previously evaluated chromosomes. Overall, the proposed method has the advantage of providing a better solution than the k-means-based method at the expense of a higher execution time and increased difficulty of achieving parallelization of the execution.

### 3.8. Deploying the System in a Cloud Solution

Another proposed concept is a generic framework for allowing the deployment of the described system architecture in a cloud solution (i.e., the open-source OpenStack solution). There is similar research in this area, namely the Stack4Things solution presented in [42], but it follows the standard architecture of an OpenStack service, and it uses the cloud IaaS services to provide the basic required functionality for an IoT solution.

An interesting approach is to deploy the proposed algorithms on cloud instances and use that instance to coordinate sensor data and instances as presented in Figure 5. The instances can be managed using the Nova instance controller. Depending on the output of the algorithms, the controller can automatically provision new instances depending on the formed clusters, which handle the rules. The sensor information can be saved with block and object storage using the Cinder and Swift services, while advanced orchestration and easy redeployment can be obtained using the Heat service. The cluster manager instance, which employs the clustering algorithms, can configure the custom sensor data load balancer to store more efficiently the sensor data based on the clusters (i.e., Nova instances) that require those data.

## 4. Results

### 4.1. Simulation Set-Up

In order to test the efficacy of the proposed solution, a simulation framework was developed, and the two proposed methods were implemented and compared. The framework makes use of three parameters: the number of sensors, number of rules and number of desired clusters. Multiple tests were performed with various combinations of the three parameters and with different seed values for the random value generator function. The selected parameters and data presented in this paper represent a test case in which the obtained results are in the middle range of performance metric values. For most of the performed tests, the result variation for each metric was ±4% compared with the presented results.

The considered test simulated 12,000 sensors with 6000 rules, and the goal was to generate 36 clusters with those rules. Each sensor produced a generated number of data units, which was a random value with a Gaussian distribution. The formula for generating the value was 500+300∗g, where *g* is a pseudo-random Gaussian distributed number with a mean of 0.0 and a standard deviation of 1.0. The range for the volume of data produced by a sensor was between 1 and around 800 d.u. Each rule received data from between 1 and 11 sensors, as shown in Figure 6.

Because this is not a straightforward approach for applying an algorithm, Figure 7 and Figure 8 provide more context and offer a perspective on how the information is generated and associated to sensors and rules. The distribution of data units among sensors and the sensor-rule association were performed in order to simulate a real sensor environment as closely as possible.

Figure 7 presents the computed data volume required by a rule, depending on the number of sensors associated with that rule. For example, there was a rule that had 11 sensors associated with it, and it required a total of 5630 d.u. to process it. In Figure 8, the number of rules that need to process different ranges of data volumes is shown (e.g., 356 rules need to process a data volume in the range (1146, 1336] d.u.).

### 4.2. Experimental Results

Each of the proposed algorithms was executed on the same generated test case, and the results were compared depending on the data volumes that needed to be processed by each cluster of rules. In the evaluation process, we used the two performance metrics described in Section 3.4. The goal was to have a low makespan and a low load imbalance.

The k-means-based algorithm was configured to initially generate only four clusters. Because the features used by the k-means method were the number of sensors and the data volume processed by each rule, each of the four clusters had grouped together rules with similar features. The total data volume required to be processed by each initial cluster was 1,645,684 d.u., 2,887,941, d.u., 2,738,362 d.u. and 1,350,257 d.u. As can be observed, the two middle clusters had to process around twice as many data compared with the other two. The first operation performed on the clusters by the k-means-based algorithm was to split the middle clusters, resulting in a total of six clusters with data volumes of 1,645,684 d.u., 1,531,453 d.u., 1,520,189 d.u., 1,350,257 d.u., 1,470,455 d.u. and 1,460,521 d.u. Each of these six clusters was split into six parts to reach the desired 36 clusters. The minimum data volume that needed to be processed by a cluster was 240,164 d.u., and the maximum (i.e., the makespan) was 289,776 d.u., resulting in a load imbalance of 49,612 d.u., which represented 17% of the makespan. This might not seem like a good result, but considering that traditional task scheduling could not be applied given the sensor-rule constraints, the proposed method was very fast and provided an adequate solution.

For an initial comparison, there were 200 random solutions generated, and the best one among them was selected by evaluating it with the fitness function of the genetic algorithm. When using the makespan as a fitness criterion, the best solution had a minimum data volume of 219,256 d.u. and a maximum of 290,519 d.u., resulting in a load imbalance of 71,263 d.u., which represented 24% of the makespan. When using the load balance as a fitness criterion, the best solution had a minimum data volume of 235,601 d.u. and a maximum of 300,391 d.u., resulting in a load imbalance of 64,790 d.u., which represented 21% of the makespan. The proposed k-means-based algorithm performed better compared with the two solutions selected among 200 random solutions using the two performance metrics. It also ran in a fraction of the time compared with the time it took to generate and evaluate 200 candidate solutions.

The tested genetic algorithm used the parameters and functions described in Section 3.7. The formula for computing the fitness depends on the number of sensors associated with each rule, the range of d.u. generated by the sensors and the considered criterion (i.e., the makespan or the load balance). Let min be the minimum data volume that needs to be processed by a cluster and max be the maximum data volume that needs to be processed by a cluster. For the presented test case, the chromosome fitness when trying to minimize the makespan was computed as $10^{8}/\left(max+1\right)$, while the chromosome fitness when trying to balance the load was computed as $10^{6}/\left(\left(max-min\right)+1\right)$. This was carried out to obtain more human-readable values for the fitness. The evolution of the results obtained by the genetic algorithm using the makespan for the fitness is presented in Figure 9, and the ones using the load balance are shown in Figure 10. For both graphs, higher values mean a better solution because the fitness is inversely proportionate with the makespan and the load imbalance. In order to better compare the results, the fitness of the k-means-based algorithm and the average fitness obtained from the initial population of the genetic algorithm were added to the charts. The average population fitness was determined by generating the initial population for the genetic algorithm and then calculating the fitness of each candidate solution. Finally, the average of the fitness values was calculated, and that resulting value represented the average population fitness.

The obtained fitness values are shown in Table 1. When trying to improve the makespan, the proposed k-means-based algorithm was 8% more efficient than the average population fitness, while the proposed genetic algorithm provided an efficiency increase of 17%. Regarding improving the load balance, the differences were more significant. The k-means-based approach was almost twice as good compared with the average population fitness, while the genetic algorithm was more than 17 times better.

How the best fit candidate solution evolved with each iteration of the genetic algorithm, in terms of the data volume that was assigned to each cluster, can be observed in Figure 11 and Figure 12. The largest data volume that needed to be processed by a cluster started from 290,519 d.u. for the GA for the makespan and from 300,391 d.u. for the GA with the load balance. The final obtained results were 266,556 d.u. and 263,889 d.u., respectively. The lower the data volume was, the better the result. An interesting aspect is the fact that the GA with the load balance obtained a lower value for the largest data volume to be processed by the cluster while also obtaining a load imbalance (in terms of the associated data volumes) of only 547 d.u. compared with 25,832 d.u. for the GA with the makespan.

An overall view of the impact of the proposed methods can be observed in Figure 13, which presents the data volumes that were assigned to be processed by each of the 36 clusters depending of the employed algorithm: a GA with the makespan for fitness (best chromosome from the final iteration), a GA with the load balance for fitness (best chromosome from the final iteration), the k-means adaptation algorithm, a GA with the makespan for fitness (best chromosome from the initial iteration) and a GA with the load balance for fitness (best chromosome from the initial iteration). The final result obtained by the GA with load balance for fitness clearly provided the best results, and it was best in most of the tested scenarios with different numbers of sensors, rules and clusters.

## 5. Discussion

The research regarding the two proposed methods presented in this paper expands on the task scheduling problem by considering an extra layer regarding the input data. In the traditional approach, the problem is summarized as having a number of tasks that need to be processed as efficiently as possible in a heterogeneous environment. If only considering the sensor data, a task can be seen as processing that data. The issue is that the sensor data must be evaluated by rule engines, which check each new piece of data against a series of rules. Therefore, traditional methods are not directly applicable. The two new methods proposed in this paper (i.e., the k-means-based clustering and the customization of the genetic algorithm) take into account this extra layer of complexity. They also provide the means to distribute the computation so that the data are transferred for processing only where needed, thus ensuring that less information goes through the network and the rules are evaluated in a shorter time. The two methods can be used in a broader context, such as having task result aggregators, which are equivalent to having rules that aggregate or process information from different specific sensors.

The main scenario in which the proposed methods can be used is if there is a known number of instances available to process sensor data. In this case, the clustering methods use as input the number of instances, and they produce a good rule-to-cluster association for handling the data. This is one of the reasons why the k-means algorithm is appropriate for solving this method. The proposed k-means solution is adapted to first generate a fixed small number of clusters and afterward split the clusters to obtain the desired number. An interesting variation is to consider a variable number of clusters and to evaluate the quality of the solution at each split. This way, using a slightly smaller number of clusters, a solution with a better performance metric might be obtained. This approach is inspired by hierarchical k-means methods, which allow the determination of the optimal number of clusters.

The proposed architecture stores all the sensor data in a global distributed storage system, which allows the correlation of data from multiple sources. Instead of having multiple local processing instances, cost efficiency is increased by using a global pool of computing instances. On the other hand, using this approach has the disadvantage of ensuring data privacy and security, but this is somewhat similar to the way that cloud providers ensure these aspects. In addition, some entities may not want their sensor data to be shared and used by artificial intelligence methods to find data or even behavioral patterns. If the solution is used for a smart city or region, and it is governed by the same entity, then the data privacy issue becomes mute.

In terms of where the data processing is performed, there are three main directions: edge computing, fog computing and cloud computing. Edge computing means that the processing is performed directly on the devices attached to the sensors. Fog computing is processing in a network close to the edge but somewhere between the edge and the cloud. Finally, cloud computing means handling the data in a cloud solution, which has access to information from a large amount of sensors. Each of them has advantages and disadvantages, and some solutions employ all three [43,44]. The current paper focuses on a cloud solution in order to have a broader view of the underlying sensor networks. Because all the data are considered, the best decision can be made but at the expense of an increased computing time. Considering a scenario with a high number of sensor networks, the costs to perform these computations are similar to the costs in the case of fog and edge computing. Instead of having centralized processing, there is processing on each fog computing node and on each edge computing node. In order to decrease costs and processing times, the rule engine system can be designed to perform processing at all three levels (edge, fog and cloud). The sensor data go to the edge level, where some local rules are triggered. Then, the relevant data go to the fog level, where other rules are used. Finally, the cloud rules perform the processing and trigger events. This method has the advantage of making quick real-time decisions at the edge level while considering the big picture at cloud level. The two proposed methods can be applied best at the fog and especially cloud levels because they thrive when a limited amount computing instances are available to process a large amount of sensor data.

Looking at the proposed cloud solution architecture from Figure 5, a mandatory discussion regards how exactly the data flow is configured and reconfigured depending on changes in the environment (e.g., new sensors are added or removed). The decision to reconfigure the data flow must be made by the cluster manager instance. The proposed algorithms can periodically run and re-evaluate the load of each cluster (i.e., processing instance) and, if needed, trigger a rule migration from one cluster to another or the creation of new clusters or instances or the destruction of existing ones. By using this approach, a high degree of scalability was achieved. The execution of the k-means-based algorithm can be easily parallelized and therefore ensure swift reaction to changes in the system, while the custom genetic algorithm can be used when major changes occur in the environment.

The issues with the proposed system and the two proposed methods were discussed throughout the paper. The efficacy was demonstrated through simulation, which tried to mimic a real multi-sensor network environment. It is highly unlikely that the system will be tested in a real-world environment due to the high amount of sensors needed. However, it can be tested for the sensor data from a real smart home, and those data can be extrapolated to simulate a smart neighborhood. Before this can be tested, the next research step is to deploy the proposed cloud solution in an in-house OpenStack cloud. The system will be generalized for handling any type of task with processing constraints. This generic solution will be offered as a PaaS service for developers, and it will also include an SaaS extension on the Horizon service for monitoring the proposed service.

## Figures and Tables

**Figure 1 sensors-23-01543-f001:**
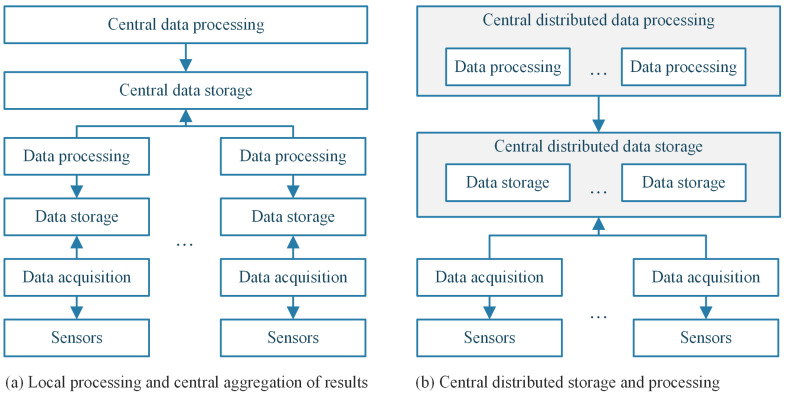
Two architectures for processing sensor data. (**a**) The sensor acquisition, storage and processing are performed locally, and the processed results are sent to the central data storage component, which is accessed by the central data-processing component for further data analysis. (**b**) The components that perform the data acquisition store the information in central distributed data storage, which is accessed by distributed data-processing components.

**Figure 2 sensors-23-01543-f002:**
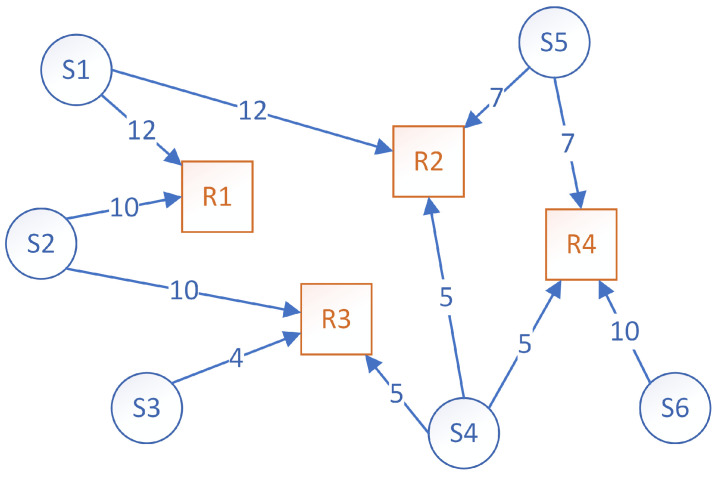
Example of the connections between six sensors (*S*) and four rules (*R*). Each sensor, represented by the round vertices, produces the amount of data specified on the graph edge, and those data have to be processed by the rules, represented by square vertices.

**Figure 3 sensors-23-01543-f003:**
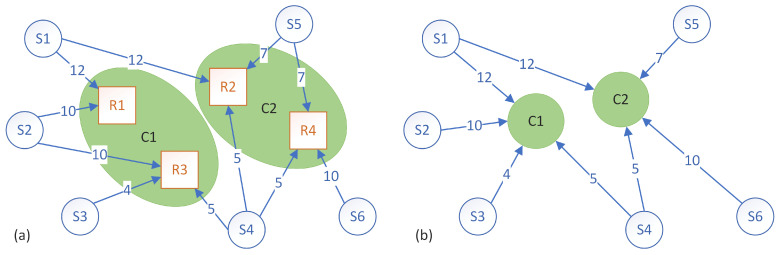
Example of clustering rules for the graph from Figure 2. (**a**) There are two clusters: $C_{1}$ with rules $R_{1}$ and $R_{3}$ and cluster $C_{2}$ with rules $R_{2}$ and $R_{4}$. (**b**) The formed clusters receive the data only once from the connected sensors (*S*).

**Figure 4 sensors-23-01543-f004:**
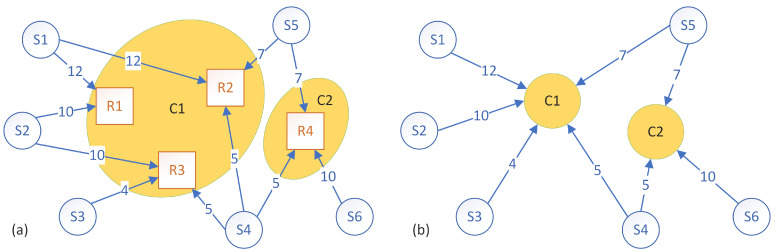
Example of clustering rules for the graph from Figure 2. (**a**) There are two clusters: $C_{1}$ with rules $R_{1}$, $R_{2}$ and $R_{3}$ and cluster $C_{2}$ with rule $R_{4}$. (**b**) The formed clusters receive the data from the connected sensors (*S*) only once.

**Figure 5 sensors-23-01543-f005:**
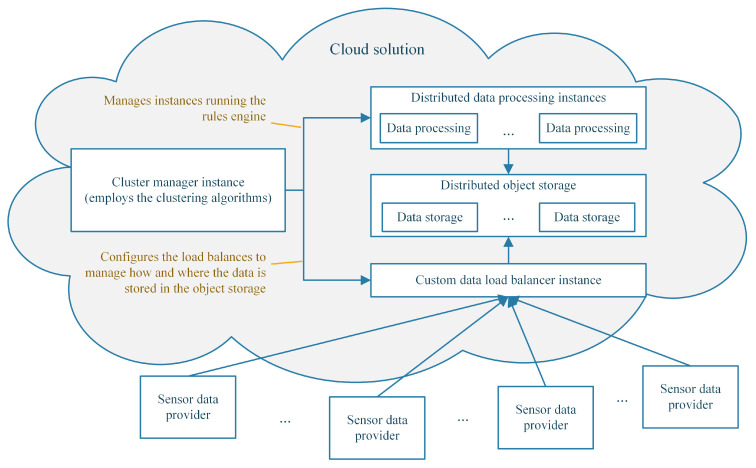
Proposed cloud solution architecture.

**Figure 6 sensors-23-01543-f006:**
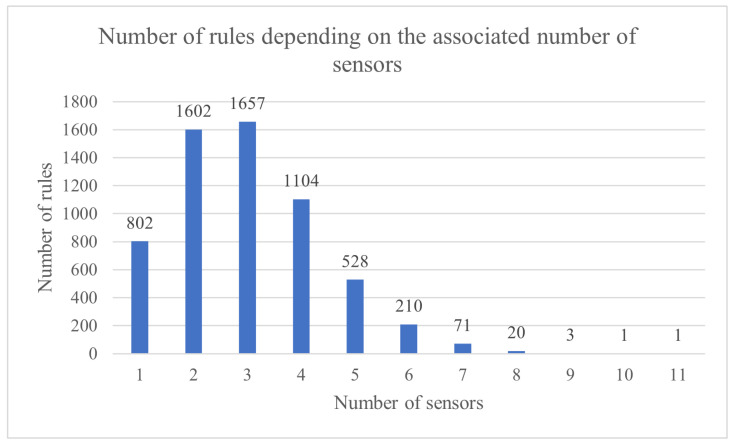
Number of rules depending on the associated number of sensors.

**Figure 7 sensors-23-01543-f007:**
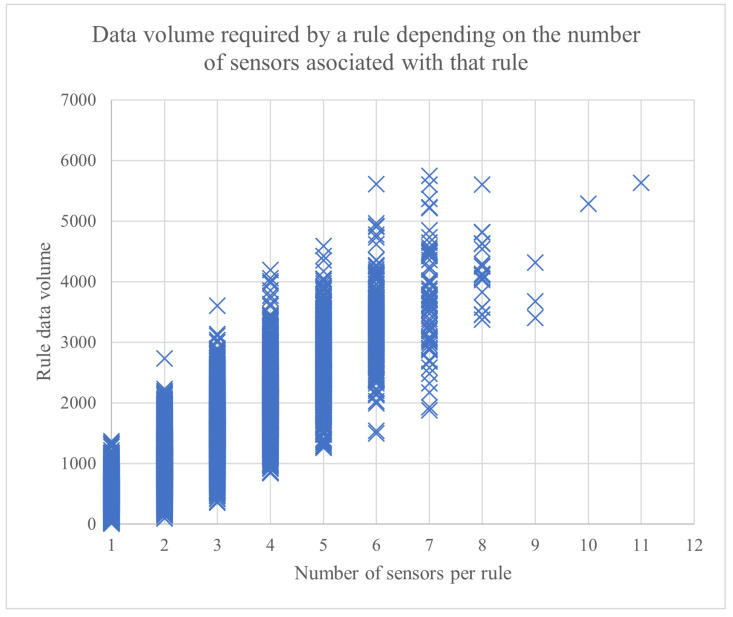
The data volume required by a rule, depending on the number of sensors associated with that rule.

**Figure 8 sensors-23-01543-f008:**
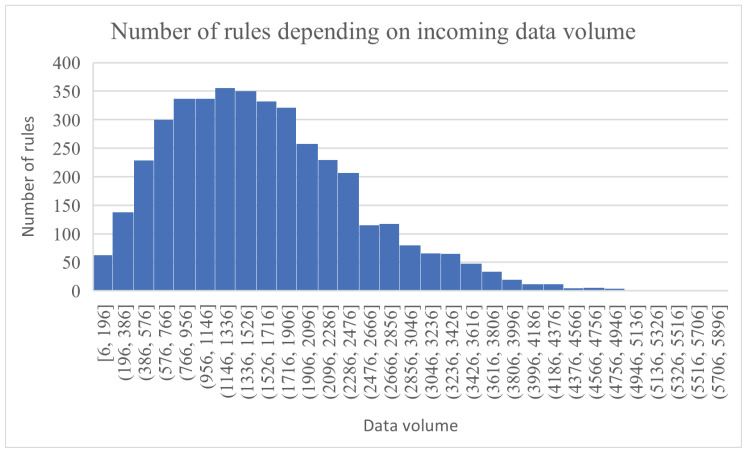
The number of rules whose incoming data from the sensors are in a specific range.

**Figure 9 sensors-23-01543-f009:**
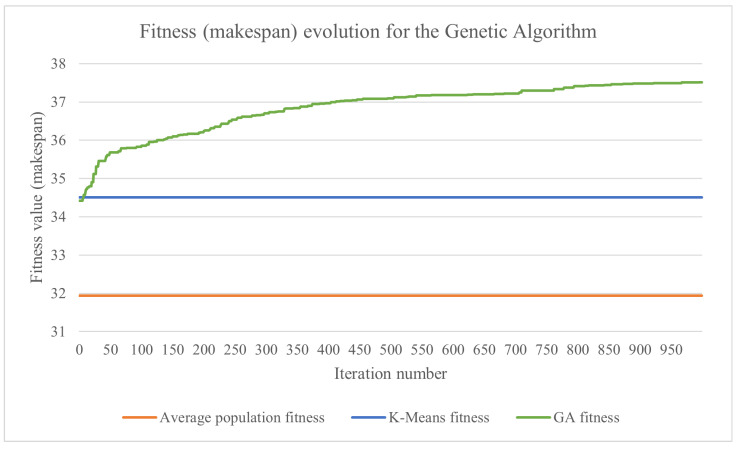
The evolution of the best chromosome’s fitness for a genetic algorithm that uses the makespan as the fitness function, with the average population fitness and k-means fitness for reference.

**Figure 10 sensors-23-01543-f010:**
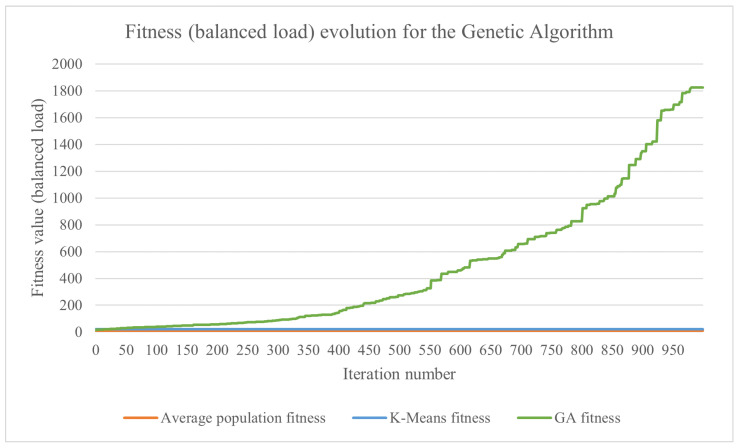
The evolution of the best chromosome’s fitness for a genetic algorithm that uses the load balance as the fitness function, with the average population fitness and k-means fitness for reference.

**Figure 11 sensors-23-01543-f011:**
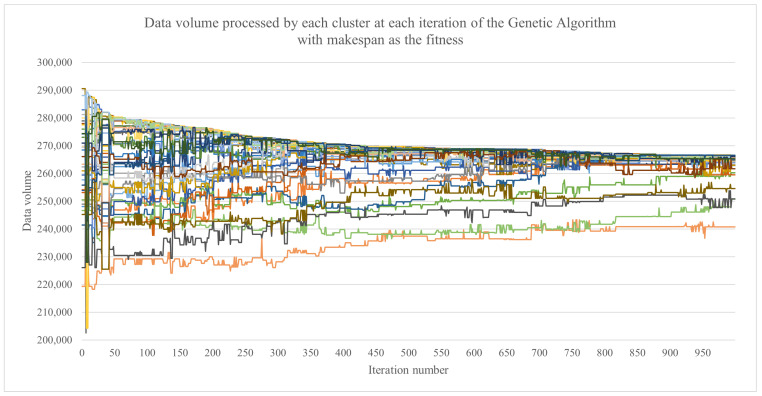
The evolution of the data volume that was assigned to each cluster, represented by the best fit candidate solution at each iteration of the genetic algorithm when using the makespan as the fitness function.

**Figure 12 sensors-23-01543-f012:**
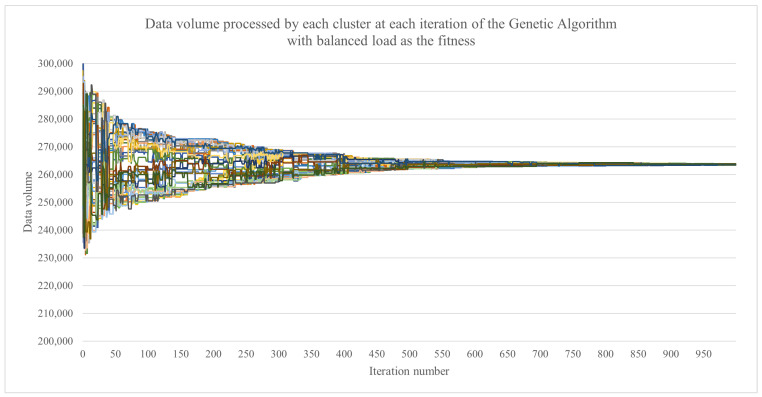
The evolution of the data volume that was assigned to each cluster, represented by the best fit candidate solution at each iteration of the genetic algorithm the load balance as the fitness function.

**Figure 13 sensors-23-01543-f013:**
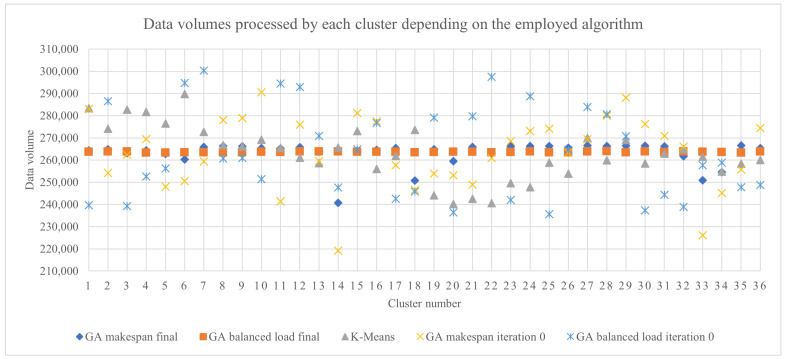
The data volumes that were assigned to be processed by each of the 36 clusters, depending on the employed algorithm.

**Table 1 sensors-23-01543-t001:** Fitness values depending on the aspect that was considered when computing the fitness (i.e., makespan or load balance).

Fitness Method	Avg. Population Fitness	K-Means-Based Fitness	Genetic Algorithm Fitness
makespan	31.94	34.51	37.52
load balance	10.51	20.16	1824.82

## Data Availability

The data presented in this paper and the code for the implemented solutions are available by requesting them from the corresponding author.

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
