# Peer review of "Parallel Processing of Sensor Data in a Distributed Rules Engine Environment through Clustering and Data Flow Reconfiguration"

_sensors, 2023, doi:10.3390/s23031543_

Round 1

Reviewer 1 Report

I would like to thank the author for submitting this work for reviewing. The review comments will be structured in (a) general remarks (these comments cover the entire manuscript), and (b) specific remarks (feedback on sentence and/or word level). Some of the specific remarks will refer to specific line number (e.g. 110). The remarks can include a quote from your original manuscript to refer to a specific section.

General remarks

This paper proposed two parallel computing methods to cluster data to obtain an efficient data flow.

The manuscript is easy to read and well organized despite some grammar and typo mistakes. The structure was thought out well and the research problem is clearly stated in the abstract and the introduction. Some background information was provided. The research is supported by listing, diagrams, and results for the performance parameters.

To further improve the clarity of presentation and the quality of the context, the following suggestions could be considered:

·       Representative title: the paper title is long. The “clustering approach” which not mentioned which is the main mechanism in the paper.

·       Paper layout is not mentioned in the introduction

·       Clarify terminology: It could be helpful to provide more explicit definitions or explanations of technical terms (e.g JSON) or concepts that may be unfamiliar to readers who are not experts in the field.

·       Language: some sentences are not clear

There are a few potential weaknesses of this work that could be considered:

·       The orchestration topic is interesting, but cloud and edge resource management and orchestration is a saturated research area. Hence, the novelty of the work was not clearly justified as the literature review did not focus on clustering (particularly in the Fog computing). Providing more context about that field could help to situate the research question and the findings of the study within a larger context.

·       The author is encouraged to expand on the relevant literature review. The author could discuss the difference between the work submitted and the dynamic Network Reconfiguration in wireless network.

·       The author did not mention the procedure currently followed in processing the data with the rule’s engines and comparison to other proposed methods.

·       The research is based on centralized processing in the cloud which most researchers moved from to edge and fog computing to reduce processing complexity, resources and time. It is good to mention in the manuscript where the two approaches could be implemented; Cloud, Fog or Edge computing.

·       No comparison to traditional approaches: the author used GA and K-mean methods, but it would be useful to see how these models compare to the traditional approaches where clustering is not used. It will be interesting to see the comparison base on complexity involved. This would provide a more comprehensive understanding of the relative strengths and weaknesses of the proposed methods.

·       Justification of the methods used: The author selected GA and K-mean methods for this paper research. Providing more context about what the characteristics for these approaches that make them the best methods for clustering would help. In other words, why are these the best methods?  

·       In the methods setting, providing details about the parameters used to cluster/group the rules together are preferable.

·       As the two approaches are applicable for specific conditions, as stated “The execution of the K-Means-based algorithm can be easily parallelized and, therefore, ensuring swift reaction to changes in the system, while the custom Genetic Algorithm can be used when major changes occur in the environment.” If they were both considered, then what is the mechanism adopted by the cluster manager to decide which approach to use for reconfiguring the data flow?

Specific remarks

Grammar and typo mistakes (e.g.142-“a”, 206- “the responsibility”, 486-“a”)

214-215 check sentence, is there a mistake?

Figure 10 includes the value for average population fitness. The authors didn’t explain how this has been calculated.

Reviewer 2 Report

The title of the paper is: Using Data Flow Reconfiguration to Improve the Parallel Processing of a Large Amount of Sensor Data in a Distributed Rules Engine Environment.The quality of the paper is good, and it is timely. It would be a good addition to the journal, provided the authors address these concerns. The paper can be accepted with major changes.

Suggested Changes 

1. In the abstract, " These methods allow for a seamless increase f the number of sensors in the environment by making smart use of the available resources," you should define this with a quantative value like 34%

2. "A solution for cloud deployment while ensuring scalability and adaptation to the sensor-rules particularities of the system"—can you explain this contribution ?and how much scaling can be done using your proposed method.

3. The organisation of the paper should be included after the introduction .

4. I can't see the related literature section. Why have you not included this section? Do add a table in this section 

5. Why are some words italicized?

6. Please explain acronyms when they are first used.

7. Watch your word order and syntax.

8. If authors think that the pseudocode of the setup phase clarifies the process, its presentation must be enhanced if possible.

9. There are some long paragraphs, the authors can divide them into shorter ones.

10. there should be a related work section, in which the description of each protocol must be placed .

11. Please explain the problems with your own solution and give more information about what needs to be done in the future.

12. suggest to add these two references to increase the impact of the paper 1. https://doi.org/10.3390/s22103809

https://doi.org/10.3390/drones6080193 

Reviewer 3 Report

The paper aims to improve the parallel processing of a large volume of sensor data in a distributed rules engine environment through data flow reconfiguration. Comments and suggestions are given as follows. 

Line 196, as the article put, each edge represents the volume of information (expressed in data units). It is supposed to give an equation to indicate the relationship between volume and data units.

Line 200, as the article put, this can easily be made possible 200 if certain rules are set to also act as sensors (i.e., produce data), what if any circle among sensors and rules affects the proposed method? 

Concerning about the cost of managing the clusters, by what way do the configuration of clusters be released to sensors and rules?

In case of a sensor being added or deleted, how to adjust clustering method? how about the performance of adjusting.

Concerning about K-means algorithm, it is supposed to analyze the selection of the optimal parameter K, which is put as the number of desired clusters (p) in Line 269. 

Round 2

Reviewer 2 Report

All my comments have been addressed so i would like to accept the manuscript in the current form .

Author Response

Thank you very much for all the feedback.

Reviewer 3 Report

According to the author's response to the comment of Line 200That statement was moved to the Discussion section of the paper and there was elaborated this situation. It is suggested to insert the statement and explanation straightly around Line 200, instead of Discussion section. The same situation is followed by the response to comments of point 3 and 4. 
